# Offline Policy Comparison with Confidence: Benchmarks and Baselines

**Anurag Koul**
School of EECS
Oregon State University
`koula@oregonstate.edu`

**Mariano Phielipp**
Intel Labs
`mariano.j.phielipp@intel.com`

**Alan Fern**
School of EECS
Oregon State University
`afern@oregonstate.edu`

## Abstract

Decision makers often wish to use offline historical data to compare sequential-action policies at various world states. Importantly, computational tools should produce confidence values for such offline policy comparison (OPC) to account for statistical variance and limited data coverage. Nevertheless, there is little work that directly evaluates the quality of confidence values for OPC. In this work, we address this issue by creating benchmarks for OPC with Confidence (OPCC), derived by adding sets of policy comparison queries to datasets from offline reinforcement learning. In addition, we present an empirical evaluation of the *risk versus coverage* trade-off for a class of model-based baseline methods. In particular, the baselines learn ensembles of dynamics models, which are used in various ways to produce simulations for answering queries with confidence values. While our results suggest advantages for certain baseline variations, there appears to be significant room for improvement in future work.

## 1   Introduction

Given historical data from a dynamic environment, how well can we make predictions about future trajectories while also quantifying the uncertainty of those predictions? Our main goal is to drive research toward a positive answer by encouraging work on a specific prediction problem, *offline policy comparison with confidence (OPCC)*.

OPCC involves using historical data to answer queries that each ask for: 1) a prediction of which of two policies is better for an initial state and horizon, where the policies, state, and horizon can be arbitrarily specified, and 2) a confidence value for the prediction. While here we use OPCC for benchmarking uncertainty quantification, it also has utility for both decision support and policy optimization. For decision support, a farm manager may want a prediction for which of two irrigation policies will best match season-level crop goals. A careful farm manager, however, would only take the prediction seriously if it comes with a meaningful measure of confidence. For policy optimization, we may want to search through policy variations to identify variations that confidently improve over others in light of historical data.

Offline reinforcement learning (ORL) [40], both for policy evaluation and optimization, offers a number of techniques relevant to decision support and OPCC in particular. One of the key ORL challenges is dealing with uncertainty due to statistical variance and limited coverage of historical data. This recognition has led to rapid progress in ORL, yielding different approaches for addressing

Offline Reinforcement Learning Workshop at Neural Information Processing Systems, 2022

uncertainty, e.g. pessimism in the face of uncertainty [36, 3, 30, 64] or regularizing policy learning toward the historical data [37, 54, 19, 34]. However, there has been very little work on directly evaluating the uncertainty quantification capabilities embedded in these approaches. Rather, overall ORL performance is typically evaluated, which can be affected by many algorithmic choices that are not directly related to uncertainty quantification. A major motivation for our work is to better measure and understand the underlying uncertainty quantification embedded in popular ORL approaches for offline policy evaluation (OPE).

**Contribution.** The first contribution of this paper is to develop benchmarks (Section 4) for OPCC derived from existing ORL benchmarks and to suggest metrics (Section 3.3) for the quality of uncertainty quantification. Each benchmark includes: 1) a set of trajectory data $D$ collected in an environment via different types of data collection policies, and 2) a set of queries $Q$, where each query asks which of two provided policies has a larger expected reward with respect to a specified horizon and initial states. Note that our OPCC benchmarks are related to recent benchmarks for offline policy evaluation (OPE) [17], which includes a policy ranking task similar to OPCC. That work, however, does not propose evaluation metrics and protocols for measuring uncertainty quantification over policy rankings. Further, our query sets $Q$ span a much broader range of initial states than existing benchmarks, which is critical for understanding how uncertainty quantification varies across the wider state space as it relates to the trajectory data $D$.

Our second contribution is to present a pilot empirical evaluation (Section 5) of OPCC for a class of approaches that use ensembles as the mechanism to capture uncertainty, which is one of the prevalent approaches on ORL. This class uses learned ensembles of dynamics and reward models to produce Monte-Carlo simulations of each policy, which can then be compared in various ways to produce a prediction and confidence value. Our results for different variations of this class provide evidence that some variations may improve aspects of uncertainty quantification. However, overall, we did not observe sizeable and consistent improvements from most of the considered variations. This suggests that there is significant room for future work aimed at consistent improvement for one or more of the uncertainty-quantification metrics.

The benchmarks and baselines are made publicly[1] available with the intention of supporting community expansion over time.

## 2 Background

We formulate our work in the framework of Markov Decision Processes (MDPs), for which we assume basic familiarity [56]. An MDP is a tuple $M = (S, A, P, R)$, where $S$ is the state space, $A$ is the action space, and $P(s'|s, a)$ is the first-order Markovian transition function that gives the probability of transitioning to state $s'$ given that action $a$ is taken in state $s$. Finally, $R(s, a)$ is potentially a stochastic reward function, which returns the reward for taking action $a$ in state $s$.

In this work, we focus on decision problems with a finite horizon $h$, where action selection can depend on the time step. A non-stationary policy $\pi(s, t)$ is a possibly stochastic function that returns an action for the specified state $s$ and time step $t \in \{0, \ldots, h - 1\}$. Given an MDP $M$, horizon $h$, and discount factor $\gamma \in [0, 1)$; the value of a policy $\pi$ at state $s$ is denoted by $V_M^\pi(s, h) = \mathbb{E}\left[\sum_{t=0}^{h-1} \gamma^t R(S_t, A_t) \middle| S_0 = s, A_t = \pi(S_t, t)\right]$, where $S_t$ and $A_t$ are the state and action random variables at time $t$. It is important to note that we gain considerable flexibility by allowing for non-stationary policies. For example, $\pi$ could be an open-loop policy or even a fixed sequence of actions, which are commonly used in the context of model-predictive control [59]. Further, we can implicitly represent the action value function $Q^\pi(s, a, h)$ for a policy $\pi$ by defining a new non-stationary policy $\pi'$ that takes action $a$ at $t = 0$ and then follows $\pi$ thereafter, which yields $V_M^{\pi'}(s, h) = Q_M^\pi(s, a, h)$. For this reason, we will focus exclusively on comparisons in terms of state-value functions without loss of generality.

---

[1]Benchmark and baselines: `https://github.com/opcciclr`

# 3 Offline Policy Comparison with Confidence

In this section, we first introduce the concept of policy comparison queries, which are then used to define the OPCC learning problem. Finally, we discuss metrics used in our OPCC evaluations.

## 3.1 Policy Comparison Queries

We consider the fundamental decision problem of predicting the relative future performance of two policies, which we formalize via *policy comparison queries (PCQs)*. A PCQ is a tuple $q = (s, \pi, \hat{s}, \hat{\pi}, h)_M$, where $s$ and $\hat{s}$ are arbitrary starting states, $\pi$ and $\hat{\pi}$ are policies, $h$ is a horizon, and $M$ is a MDP. The answer to a PCQ is the truth value of $V_M^\pi(s, h) < V_M^{\hat{\pi}}(\hat{s}, h)$. That is, a PCQ asks whether the $h$-horizon value of $\hat{\pi}$ is greater than $\pi$ when started in $\hat{s}$ and $s$ respectively.

As motivated in Section 1, PCQs are useful for both human-decision support and automated policy optimization. For example, if a farm manager wants information about which of two irrigation policies, $\pi$ and $\hat{\pi}$, will result in the best future crop yield given the environment state $s$, then the corresponding PCQ would be $(s, \pi, s, \hat{\pi}, h)_M$. Alternatively, the manager may be interested in whether a policy $\pi$ is better suited to an environmental state $s$ or $\hat{s}$, which is captured by the PCQ $(s, \pi, \hat{s}, \pi, h)_M$. In addition, PCQs can be used as the basis for the classic policy improvement step of policy iteration [56]. We elaborate our discussion in Appendix A.3.

## 3.2 Learning to Answer PCQs with Confidence

Given an accurate generative model of the environment MDP $(M)$, a PCQ $(s, \pi, \hat{s}, \hat{\pi}, h)_M$ can be answered via Monte Carlo trajectory sampling to estimate $V_M^\pi(s, h)$ and $V_M^{\hat{\pi}}(\hat{s}, h)$. Further, the confidence in the answer can be arbitrarily improved by increasing the number of sampled trajectories. In this work, we do not assume an environment MDP $(M)$, but instead are provided with an offline data set of environment trajectories produced by one or more unknown behavior policies. We will denote this dataset by $\mathcal{D} = \{(s_i, a_i, s_i', r_i)\}$ where each tuple corresponds to an observed transitions from state $s_i$ to state $s_i'$ after taking action $a_i$ and receiving reward $r_i$. For notational simplicity, we will omit $M$ from PCQs and value functions; assuming the MDP is implicit through dataset $\mathcal{D}$.

Given a dataset $\mathcal{D}$ we would like to learn a model for predicting answers to PCQs from a query space $\mathcal{Q}$. Here, $\mathcal{Q}$ may assert application-specific restrictions on states and policies involved in PCQs. A fundamental challenge is that the coverage of $\mathcal{D}$ will not necessarily be representative of the dynamics and rewards relevant to answering all queries in $\mathcal{Q}$. Thus, if query answers are being used to inform important decisions, then it is critical for each answer to come with a meaningful measure of confidence that accounts for data coverage and statistical variance. Dealing with this uncertainty is also a core challenge for general offline RL [40], which has lead to a number of approaches for addressing it. However, there is little direct evaluation of the uncertainty-handling components.

The above motivates the *OPCC learning problem*, which provides a dataset $\mathcal{D}$ and desired constraints on the query space $\mathcal{Q}$. The learner should output a model $w = (f, c)$ composed of: 1) a query prediction function $f : \mathcal{Q} \to \{0, 1\}$, which returns a binary answer for any query in $\mathcal{Q}$, and 2) a confidence function $c : \mathcal{Q} \to [l, u]$ that maps queries in $\mathcal{Q}$ to a confidence value within a bounded interval. Given a query $q$, the intent is for larger values of $c(q)$ to indicate a higher confidence in the prediction $f(q)$. Note that we do not attach any predefined semantics to the values of $c(q)$ to allow for flexibility of potential solutions. Rather, we focus on defining metrics for directly evaluating the quality of uncertainty quantification provided by $w$. If desired, various methods can be used after learning to calibrate the confidence values of $c$ to meaningful scales (e.g. [41, 49]. Section 5 discusses possible learning approaches and the baselines evaluated in this paper.

## 3.3 Evaluation Metrics

In the following, we introduce our metrics and a brief about their intra-relation could be found in Appendix B. Our metrics are designed to evaluate both the query answer (or rank) and confidence; in contrast to OPE metrics used by [67, 17, 53, 28].

**Area under risk-coverage curve (AURCC).** In selective classification, the aim is to reduce prediction errors by allowing a predictor to abstain from a prediction if the confidence is below a threshold. *The quality of confidence values is thus related to how well they result in abstaining when the prediction*

*would have been incorrect*. This idea is formalized via *risk-coverage curves (RCCs)* by [71] and is outlined below.

Let $\mathcal{L}(q, \hat{y})$ be a loss function for predicting $\hat{y}$ for query $q$, e.g. 0/1 loss. Given a test set of queries $Q = \{q_1, \ldots, q_N\}$, a model $w = (f, c)$, and confidence threshold $\tau$, the *coverage* is the fraction of test queries with confidence at least $\tau$. The *selective risk* is the average loss of $f$ over the covered queries. Formally, the *coverage* and *selective risk* are respectively define by

$$cov(w, Q, \tau) = \frac{1}{|Q|} \sum_{q \in Q} I[c(q) \geq \tau] \quad \text{and} \quad r(w, Q, \tau) = \frac{\sum_{q \in Q} I[c(q) \geq \tau]\mathcal{L}(q, f(q))}{\sum_{q \in Q} I[c(q) \geq \tau]}$$

where $I$ is the binary indicator function. Thus, each possible threshold corresponds to a risk-coverage operating point $(r(w, Q, \tau), cov(w, Q, \tau))$. An RCC [71] is simply the risk versus coverage curve of these operating points when sweeping through possible thresholds. The curve starts at the point $(0, 0)$, since the risk is 0 at zero coverage, and ends at $(r_f, 1)$, where $r_f$ is the risk of $f$ evaluated on all of $Q$. In order to provide a single measure of the RCC quality, we aggregate across all thresholds to compute the *Area Under the RCC* (**AURCC**). Since lower risk is preferred, we consider a lower AURCC to indicate better confidence estimation.

**Reverse Pair Proportion (RPP).** Our second selective-classification metric is *reverse pair proportion (RPP)* from [69]. The main idea is that the ordering of confidence values for a pair of queries should reflect the relative prediction loss for those queries. RPP measures how often the confidence value ordering conflicts with the relative losses across all pairs of queries. In particular, a conflict occurs when the loss of $q_1$ is less than $q_2$, but we are more confident about $q_2$ than $q_1$. The $RPP(w, Q) = \frac{1}{|Q|^2} \sum_{q_1, q_2 \in Q} I[l(q_1) < l(q_2), c(q_1) < c(q_2)]$ is just the fraction of such conflicts; where $l(q) = L(q, f(q))$ is the loss of $f$ on $q$.

**Coverage Resolution ($\text{CR}_K$).** Finally, we introduce a new metric on just the confidence function $c$. A practical difference between different confidence functions is the resolution of values that they output in practice. For example, given a set of queries $Q$, one confidence function $c_1$ may result in only three distinct coverage values $cov(w, Q, \tau)$ across all thresholds, while another confidence function $c_2$ results in $|Q|$ distinct coverage values. All else being equal $c_2$ is the preferable function, since it provides a higher level of resolution with respect to abstention/coverage rates. We measure this via *coverage resolution at* $K$, denoted $CR_K$. To compute $CR_K$ for $w = (c, f)$ and query set $Q$, the coverage interval $[0, 1]$ is partitioned into $K$ equal bins and we return the fraction of bins which contain $cov(w, Q, \tau)$ for some threshold $\tau$. By increasing $K$ we get a finer grained distinction in coverage resolution.

# 4  OPCC Benchmark Construction

In this section, we briefly describe our choice of environments and approach for constructing testing query sets for OPCC benchmark. An extended summary of the benchmark and construction steps can be found in Appendix A.

**Environments.** To support easier adoption of our benchmarks, we selected seven environments and corresponding datasets that are currently used in offline RL research. We describe them as follows:

- **Maze2d (4 environments):** They were introduced in D4RL [18] and comprise of 2d mazes of different complexities: *open, u-maze, medium,* and *large* as illustrated in Figure 2 (Appendix). We use the datasets provided by D4RL, which we refer to as *"1M"* due to the datasets each having 1 million state transitions.

- **Gym-Mujoco (3 environments):** We consider three locomotion-based environments from OpenAI Gym [2]: *HalfCheetah, Walker2d, and Hopper.* These are shown in Figure 3 (Appendix). For each environment, we use the corresponding D4RL [18] datasets that include behavior trajectories of varying qualities. This includes *"random, medium, medium-replay, medium-expert, and expert".*

**Query Set Construction.** For each environment, we create a set of PCQs with ground truth answers. This is done by first generating a set of diverse behavior policies. We choose to have 5 policies for each maze2d environment and 10 policies for each gym-mujoco environment. Secondly, we generate

a large set of initial states by running these policies along with a random policy. This gives us seed states for PCQs that go beyond initial-state distribution of environments. Thereafter, for each horizon $h \in \{10, 20, 30, 40, 50\}$; we sample states from this set to create PCQs of of the form $(s, \pi, s, \hat{\pi}, h)$ where $s$ is a random initial state and $\pi, \hat{\pi}$ a random pair of the learned policies. In addition, we create a set of PCQs of the form $(s, \pi, \hat{s}, \hat{\pi}, h)$ in the same way, except that two random initial states are used instead of one.

# 5   OPCC Baselines

In this section, we describe the class of baselines that will be made available with the benchmarks and included in our pilot experiments (Section 6). Recall that each baseline must provide a prediction function $f$ and confidence function $c$ that are derived from the dataset $\mathcal{D}$. Perhaps the most natural approach for $f$ to answer a PCQ $(s, \pi, \hat{s}, \hat{\pi}, h)$ is to estimate and then compare the relevant values using OPE. The corresponding confidence function $c$ might then be based on the uncertainty of the value estimates.

There are at least two types of OPE approaches to consider: model-free and model-based. Model-free approaches, such as fitted Q-evaluation [14] typically learn a Q-function $Q^\pi(s, a)$ for a given policy $\pi$ that can be evaluated for any state and action. Unfortunately, each function learned by such model-free methods is valid for the single policy $\pi$ and the effective horizon used during training. Thus, answering PCQs involving other policies or horizons requires costly retraining. Since we are seeking an OPCC approach, which can be quickly applied to arbitrary policies, states, and horizons, we instead choose to use a model-based approach for our baselines.

Our baselines are variants of model-based ensemble approaches, which are one of the most common class of approaches used in model-based RL for dynamics modeling and capturing uncertainty [1, 73, 32]. Overall, our baselines all have the following structure: 1) Learn an ensemble of models $\{\hat{P}_i\}$ from $\mathcal{D}$ that each predict the dynamics and reward of the environment. 2) Use each model in the ensemble to generate estimates of the relevant PCQ values, $V^\pi(s, h)$ and $V^{\hat{\pi}}(\hat{s}, h)$, via Monte-Carlo simulation of the policies. 3) Combine the ensemble estimates to provide a prediction and confidence value.

## 5.1   Base Models

In our experiments, we consider two types of base models for forming ensembles. The first base model is the commonly use *Feed-Forward (FF)* Gaussian model, which, given the current state/observation and action as input, returns the mean and diagonal covariance matrix of a Gaussian distribution over the next state and reward. This model allows for stochastic Monte-Carlo simulations by drawing the next state from the model's Gaussian distribution at each time step. In this work, we use the same FF base-model architecture and training details as MBPO [29].

We also consider a recent base model [74] (referred to as *Auto-regressive (AR)*), which was demonstrated in some cases to improve over the output architecture of FF. Instead of generating all $n$ features of the predicted next state in a single pass, AG auto-regressively samples each feature one at a time using $n$ forward passes. In particular, to sample state feature $i$ of the next state, denoted $s_{t+1}^i$, the network receives the usual input $s_t$ and $a_t$ as well as the previously sampled state features $s_{t+1}^0, ... s_{t+1}^{i-1}$. AR then returns the mean and covariance for a Gaussian that is used to sample $s_{t+1}^i$. The intuition is that this approach may allow for representing non-Gaussian and multi-modal next-state distributions compared to the uni-modal Gaussian FF model.

## 5.2   Ensemble Learning

Model-based approaches to ORL have commonly used ensembles as an attempt to quantify uncertainty, e.g. via measures of ensemble-member disagreement [29]. We consider two choices for generating ensembles. The first choice is the standard *bootstrapping ensemble* approach, which simply trains each ensemble member using a different random weight initialization and bootstrapped dataset $\hat{\mathcal{D}}$ by sampling from $\mathcal{D}$ with replacement $|\mathcal{D}|$ times. The intent is that the combination of classic statistical bootstrapping [13] and random initialization will produce a diverse set of ensemble models.

Often, however, it is observed that the basic bootstrapping approach does not create enough diversity in an ensemble, which is counter to our motivation of representing uncertainty. For this reason, there are a number of proposals for increasing the ensemble diversity, of which, we consider just one in this work. In particular, work motivated by capturing uncertainty in ORL proposed the use of *randomized constant priors* to increase ensemble diversity [51]. For each base model, a randomized constant prior is produced, which is simply a network with random initial weights. The base model is trained as an additive component on top of this prior and the final output is the sum of the two. The intuition is that the constant prior should cause ensemble members to disagree more often in unrepresented parts of the state-space, which will provide a better measure of disagreement-based uncertainty.

### 5.3 Prediction and Confidence Values

Given a PCQ $(s, \pi, \hat{s}, \hat{\pi}, h)$ query and ensemble of size $N$ we generate a prediction and confidence by first using each ensemble member to generate, via Monte-Carlo simulation, a pair of value estimates of $V^\pi(s, h)$ and $V^{\hat{\pi}}(\hat{s}, h)$. This results in a set of $N$ value estimate pairs, denoted by $\mathcal{V} = \{(V_1, \hat{V}_1), \ldots, (V_N, \hat{V}_N)\}$. Given the set $\mathcal{V}$, we describe the three approaches we consider for producing predictions and confidences.

**Ensemble Voting (EV).** Following [11], EV simply returns a prediction for a query based on the majority vote across the ensemble of $V_i < \hat{V}_i$. The confidence score is equal to the fraction of ensemble members that agree on the majority vote (in the range [0.5,1]), but re-scaled to fall in the range [0,1].

**Paired Confidence Interval (PCI).** The PCI confidence value is computed by estimating the expected value of $V - \hat{V}$ for a random run of the learning algorithm. The mean estimate is given by $\sum_i V_i - \hat{V}_i$ and the prediction is based on the sign of this estimate. The confidence value is based on computing $\alpha$ percentile confidence intervals on the difference, denoted by $[l_\alpha, u_\alpha]$. In particular, it is equal to the largest value of $\alpha$ such that $0 \notin [l_\alpha, u_\alpha]$. Thus, a high confidence value reflects that there is strong evidence that the expected difference is either above or below zero (in agreement with the prediction). Confidence intervals are computed based on the $t$ distribution.

**UnPaired Confidence Interval (U-PCI).** This approach makes the prediction in the same way as PCI, but uses unpaired confidence intervals to compute the confidence, which should be expected to be more conservative. In particular, we compute $\alpha$ percentile confidence intervals for the means of the $V_i$ and $\hat{V}_i$ denoted respectively by $[l_\alpha, u_\alpha]$ and $\left[\hat{l}_\alpha, \hat{u}_\alpha\right]$ and let the confidence be the maximum value of $\alpha$ for which the confidence intervals do not overlap.

## 6 Experiments

Our pilot experiments explore the baseline methods on our benchmarks using the proposed metrics for OPCC. It is important to note that these experiments are not intended to identify a top performer. Rather our primary goal for these pilot experiments is to assess the adequacy of the benchmarks and metrics for future work and to establish a basic performance bar. Secondarily, we are interested to observe evidence or the lack of evidence for certain assumptions that might be drawn about the baselines from prior work.

In our experiments, unless otherwise specified the default model is an ensemble of 100 deterministic feed-forward models and uses EV for the confidence score. For brevity, the figures, tables, and analysis in the main paper are for the Gym-Mujoco environment. We note similarities/differences for Maze2d in the main paper and refer to the Appendix for figures and tables.

**Too hard or too easy?** We first assess the degree of difficulty posed by our OPCC benchmark for our baselines. Figure 1 show RCCs of our default model for different data set types (averaged across the different PCQ horizons $h$) in gym-mujoco. Table 1 report their corresponding metrics i.e. AURCC, RPP, $CR_k$, and Loss (or risk) at complete coverage.

First, we consider risk at complete coverage and find that there is no significant difference in risk across dataset type, but varies significantly across gym-mujoco environments. This shows that some environments are more challenging than others due to their underlying complex dynamics and high dimensional observation and action sizes. Also, the risk at complete coverage for maze2d

environments with a single dataset ('1m') is significantly lower than gym-mujoco. This is potentially due to data collection via a path-planning procedure leading to significant state-action space coverage. Further Medium and Umaze have very small risks without much room for risk improvement, while Large and Open appear to have room for improvement. Second, we consider how the risk varies across coverage values. In most cases, there are no thresholds that produce points within the coverage interval (0,0.5], which indicates a lack of sensitivity in that coverage range. There are typically multiple points between (0.5, and 1], though often just a few. Ideally we would hope for a more gradual degradation in risk spanning from no coverage to complete coverage. This suggests that there is significant room to improve the coverage sensitivity, especially in the range [0,0.5].

Overall, the current set of benchmarks, with the exception of 2 Maze2d environments, are not too easy and appears to offer significant room for improvement in terms of both overall risk and sensitivity of the RCCs across coverage values. Likewise, the observation that the risks achieved are significantly less than chance suggest that the benchmarks are not too hard.

**Impact of dataset type.** The different types of data sets provide different types of coverage of the system dynamics. Is there evidence that our baselines are able to distinguish among these types? Figure 1 shows that the RCCs for different datasets are quite similar for each of the gym-mujoco environment. The AURCC and RPP values in Table 1 are consistent with these observations. This could be due to the diverse coverage of queries across the state space that offer challenges for all datasets. The small variation in RCCs across dataset types could also be due to the models learned from different datasets providing similar types of generalization. It is also possible that differences between dataset types would become more prevalent for smaller versions of the datasets, which an interesting future extension to the benchmarks. Finally no significant patterns for CR in relation to data-set type are apparent, which is not surprising since CR is expected to be more heavily influenced by the type of baseline approach.

Table 1: Evaluation metrics for *dataset-type* comparison in *gym-mujoco* environments. This includes mean and confidence intervals estimates at $95\%$ confidence level for metrics corresponding to 5 (seed) dynamics trained over each dataset.

| ENV. | DATASET QUALITY | AURCC($\downarrow$) | RPP($\downarrow$) | $CR_{10}(\uparrow)$ | LOSS($\downarrow$) |
|---|---|---|---|---|---|
| HOPPER | RANDOM | $0.156 \pm 0.008$ | $0.045 \pm 0.004$ | $0.54 \pm 0.043$ | $0.273 \pm (< 0.001)$ |
| | MEDIUM | $0.133 \pm 0.001$ | $0.03 \pm 0.001$ | $0.4 \pm (< 0.001)$ | $0.26 \pm 0.002$ |
| | EXPERT | $0.152 \pm 0.002$ | $0.04 \pm 0.001$ | $0.5 \pm (< 0.001)$ | $0.284 \pm 0.002$ |
| | MEDIUM-EXPERT | $0.136 \pm 0.001$ | $0.028 \pm (< 0.001)$ | $0.4 \pm (< 0.001)$ | $0.265 \pm 0.001$ |
| | MEDIUM-REPLAY | $0.128 \pm 0.001$ | $0.012 \pm 0.001$ | $0.3 \pm (< 0.001)$ | $0.258 \pm 0.001$ |
| HALF CHEETAH | RANDOM | $0.206 \pm 0.001$ | $0.023 \pm 0.001$ | $0.3 \pm (< 0.001)$ | $0.378 \pm 0.001$ |
| | MEDIUM | $0.222 \pm 0.001$ | $0.048 \pm 0.001$ | $0.5 \pm (< 0.001)$ | $0.374 \pm 0.002$ |
| | EXPERT | $0.212 \pm 0.002$ | $0.05 \pm 0.001$ | $0.5 \pm (< 0.001)$ | $0.361 \pm 0.002$ |
| | MEDIUM-EXPERT | $0.24 \pm 0.004$ | $0.06 \pm 0.002$ | $0.6 \pm (< 0.001)$ | $0.387 \pm 0.003$ |
| | MEDIUM-REPLAY | $0.216 \pm 0.001$ | $0.04 \pm 0.001$ | $0.4 \pm (< 0.001)$ | $0.368 \pm 0.001$ |
| WALKER 2D | RANDOM | $0.067 \pm 0.001$ | $0.024 \pm 0.001$ | $0.54 \pm 0.043$ | $0.165 \pm 0.001$ |
| | MEDIUM | $0.069 \pm 0.001$ | $0.007 \pm (< 0.001)$ | $0.22 \pm 0.035$ | $0.156 \pm 0.001$ |
| | EXPERT | $0.064 \pm 0.001$ | $0.011 \pm (< 0.001)$ | $0.3 \pm (< 0.001)$ | $0.161 \pm (< 0.001)$ |
| | MEDIUM-EXPERT | $0.068 \pm (< 0.001)$ | $0.007 \pm (< 0.001)$ | $0.24 \pm 0.043$ | $0.153 \pm 0.001$ |
| | MEDIUM-REPLAY | $0.07 \pm 0.001$ | $0.005 \pm (< 0.001)$ | $0.2 \pm (< 0.001)$ | $0.161 \pm 0.001$ |

**Impact of Query Horizon.** Learned dynamics are well known to suffer from error accumulation in multi-step rollouts. This leads to the hypothesis that OPCC performance might degrade with increasing query horizons. In Table 2 and Table 13 (Appendix) we provides metrics for various horizons $h$ averaged across data-set types. As expected, we observe higher AURCCs for longer horizons, which provides positive evidence for the hypothesis. Interestingly, we observe that in most of the environments we have very low risk for short horizons. In general, we observe AURCCs for $h = 10$ or $h = 20$ are at least an order of magnitude smaller than for larger horizons across the benchmark. This suggests a possible threshold effect for OPCC with respect to increasing horizon due to error accumulation. It also suggests our current baselines are better suited for applications like reliable policy improvement with smaller horizons.

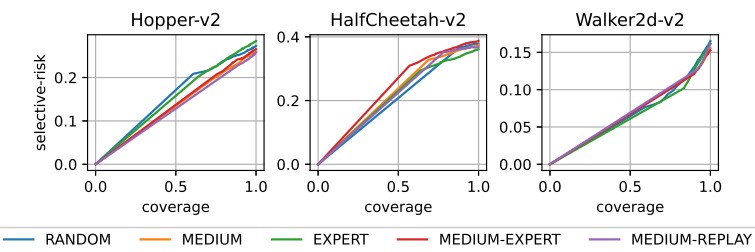

Figure 1: Selective-risk coverage curves for different *gym-mujoco* environements and *dataset types* (depicted by different colors). The x-axis spans from no(0) coverage to complete(1) coverage of queries and the y-axis is the risk for the corresponding query coverage. Each risk-coverage point is determined by varying the confidence threshold.

Table 2: Evaluation metrics for *horizon* comparison in *gym-mujoco* environments. These mean and confidnce interval(95%) estimates are over 50 samples corresponding to 5 (seed) dynamics trained over 5 different datasets.

| ENV. | HORIZON | AURCC($\downarrow$) | RPP($\downarrow$) | CR$_{10}$($\uparrow$) | LOSS($\downarrow$) |
|---|---|---|---|---|---|
| HOPPER | 10.0 | $0.017 \pm 0.002$ | $0.001 \pm (< 0.001)$ | $0.2 \pm (< 0.001)$ | $0.048 \pm 0.002$ |
| | 20.0 | $0.078 \pm 0.002$ | $0.016 \pm 0.003$ | $0.336 \pm 0.038$ | $0.17 \pm 0.003$ |
| | 30.0 | $0.146 \pm 0.004$ | $0.029 \pm 0.004$ | $0.42 \pm 0.033$ | $0.284 \pm 0.005$ |
| | 40.0 | $0.169 \pm 0.007$ | $0.038 \pm 0.006$ | $0.432 \pm 0.036$ | $0.293 \pm 0.004$ |
| | 50.0 | $0.196 \pm 0.009$ | $0.047 \pm 0.007$ | $0.516 \pm 0.049$ | $0.334 \pm 0.006$ |
| HALF CHEETAH | 10.0 | $0.077 \pm 0.004$ | $0.008 \pm 0.001$ | $0.288 \pm 0.017$ | $0.191 \pm 0.006$ |
| | 20.0 | $0.217 \pm 0.006$ | $0.041 \pm 0.005$ | $0.416 \pm 0.042$ | $0.374 \pm 0.005$ |
| | 30.0 | $0.215 \pm 0.004$ | $0.038 \pm 0.005$ | $0.404 \pm 0.038$ | $0.377 \pm 0.005$ |
| | 40.0 | $0.223 \pm 0.006$ | $0.049 \pm 0.006$ | $0.464 \pm 0.041$ | $0.368 \pm 0.005$ |
| | 50.0 | $0.277 \pm 0.008$ | $0.063 \pm 0.007$ | $0.516 \pm 0.054$ | $0.428 \pm 0.003$ |
| WALKER 2D | 10.0 | $0.011 \pm (< 0.001)$ | $0.001 \pm (< 0.001)$ | $0.22 \pm 0.016$ | $0.033 \pm 0.003$ |
| | 20.0 | $0.025 \pm 0.002$ | $0.003 \pm 0.001$ | $0.252 \pm 0.042$ | $0.077 \pm 0.004$ |
| | 30.0 | $0.059 \pm 0.001$ | $0.01 \pm 0.003$ | $0.3 \pm 0.061$ | $0.132 \pm 0.004$ |
| | 40.0 | $0.093 \pm 0.002$ | $0.017 \pm 0.004$ | $0.384 \pm 0.049$ | $0.209 \pm 0.004$ |
| | 50.0 | $0.131 \pm 0.002$ | $0.023 \pm 0.005$ | $0.392 \pm 0.06$ | $0.259 \pm 0.002$ |

**Influence of different confidence functions.** Table 3 and Table 12(Appendix) gives metrics for our three different uncertainty functions (EV, PCI, and U-PCI) averaged over data-set types and horizons. The results for AURCC and RPP both indicate evidence that the confidence interval approaches (PCI and U-PCI) have an advantage over EV. This is encouraging as it suggests considering other more sophisticated statistical testing approaches may lead to further improvement. However, the results for CR indicate that the confidence interval approaches have significantly less resolution than EV. This may lead to poorer performance for probability calibration approaches applied to PCI or U-PCI confidence scores. Further work is required to understand this decrease in resolution.

**Impact of Dynamics Choices.** There are various choices for dynamics architecture. In our work, we investigate ablated impact of design choices like size of ensemble, stochasticity, auto-regressiveness, and randomized constant priors. In Appendix D, we discuss each of these aspects in detail. As we increase the ensemble size from 10 to 100, we found weak evidence of improvement in AURCC. However, RPP and Coverage resolution tend to increase significantly. With a stochastic model outputting a normal distribution over the next observation, we don't gain significantly on any of our evaluation metrics. AR model as compared to a FF model marginally( not statistically signifincant) reduces AURCC in some cases as well as increases $CR_k$ for gym-mujoco environments only. Finally, we added randomized constant priors to our base model to encourage ensemble diversity. For maze2d environment, we observe slight improvement in AURCC for maze2d environment. whereas RPP and $CR_K$ tends to remains same. On the contrary, in the case of gym-mujoco, we generally observe a slight (but statistically insignificant) increase in AURCC , RPP, and $CR_K$.

Table 3: Evaluation metrics for *uncertainty-type* comparison in *gym-mujoco* environments. These mean and confidence interval(95%) estimates are over 50 samples corresponding to 5 (seed) dynamics trained over 5 different datasets.

| ENV. | UNCERTAINTY TYPE | AURCC($\downarrow$) | RPP($\downarrow$) | CR$_{10}$($\uparrow$) | LOSS($\downarrow$) |
|---|---|---|---|---|---|
| HOPPER | EV | $0.141 \pm 0.005$ | $0.031 \pm 0.005$ | $0.428 \pm 0.034$ | $0.268 \pm 0.004$ |
| | PCI | $0.135 \pm 0.002$ | $0.004 \pm 0.001$ | $0.2 \pm (< 0.001)$ | $0.269 \pm 0.004$ |
| | U-PCI | $0.135 \pm 0.003$ | $0.009 \pm 0.002$ | $0.216 \pm 0.014$ | $0.269 \pm 0.004$ |
| HALF CHEETAH | EV | $0.219 \pm 0.005$ | $0.044 \pm 0.005$ | $0.46 \pm 0.04$ | $0.374 \pm 0.004$ |
| | PCI | $0.191 \pm 0.002$ | $0.006 \pm 0.001$ | $0.2 \pm (< 0.001)$ | $0.373 \pm 0.004$ |
| | U-PCI | $0.196 \pm 0.003$ | $0.014 \pm 0.002$ | $0.228 \pm 0.018$ | $0.373 \pm 0.004$ |
| WALKER 2D | EV | $0.068 \pm 0.001$ | $0.011 \pm 0.003$ | $0.3 \pm 0.051$ | $0.159 \pm 0.002$ |
| | PCI | $0.078 \pm 0.001$ | $0.002 \pm 0.001$ | $0.2 \pm (< 0.001)$ | $0.16 \pm 0.003$ |
| | U-PCI | $0.076 \pm 0.001$ | $0.004 \pm 0.001$ | $0.22 \pm 0.016$ | $0.16 \pm 0.003$ |

## 7   Related Work

**Dynamics Learning in RL.** There has been much recent interest in learning deep models of dynamical systems to support model-based RL. Examples from online RL include [7, 38], which learn one-step observation-based dynamics along with extensions to ensembles [10, 6, 29, 50]. PILCO [10, 20] is another model-based RL approach that learns dynamics via Gaussian Processes [58], which are able to capture epistemic uncertainty. However, performance is primarily measured in terms of overall task performance and it is unclear how well uncertainty is actually quantified. Recent work on offline reinforcement, such as MBOP[1], MOPO [73], and MoREL [32] has also considered learning dynamics models over observations from fixed, offline data sets. These approaches incorporate uncertainty estimates in different ways (e.g. pessimistic rewards or dynamics) and all use ensembles to estimate uncertainty.

Going beyond observation-prediction model; COMBO [72], Muzero Unplugged [62], and LOMPO [57] investigate learning latent-space dynamics-models [21, 23, 22, 61, 35] for offline RL rather than learning in the observation space. In specific, LOMPO [57] learns ensemble of latent space models with constraints to have similar latent representation. Also, [70] studies representation for OPE and primarily suggests that unsupervised learning for representation helps improve policy performance.

In all the discussed dynamics learning methods, the prime focus lies in the final task performance evaluation and it's unclear how well uncertainty is actually captured by the models. Though in our work, we restrict ourselves to only observation-based models; we emphasize on better understanding of uncertainty. Also, similar to our motivation, [42] recently compares various uncertainty heuristics in model-based OPE and share various insights such as role of ensemble-size and imagination horizon length.

**Uncertainty.** Capturing uncertainty has been studied for a number of quantities relevant to RL, for example, to capture the variance of Q-values [5] , learned model-dynamics [6], and modeling data collection policies [60]. Rather than trying to cover all prior quantities and uncertainty metrics, in this paper, we have chosen to focus exclusively on uncertainty in policy comparisons (i.e. OPCC). This choice is based on the simplicity of OPCC combined with the immediate utility it has in decision making. Indeed, in many decision-making settings, we only need to policies (open- and/or closed-loop), rather than precisely estimate their values. Importantly, advancements made in more refined uncertainty estimation approaches, such as for Q-values, can be evaluated within the OPCC framework and yield advancements.

**Policy Ranking.** [25] is one of the early works that expresses the need of policy comparison with OPE under uncertainty estimation. They estimate uncertainty over OPE using the method of Uncertainty propogation [8] and show results only on discrete MDPs. Similarly, [65] also identifies the need to measure uncertainty for policies learned with limited data. In order to learn safe policies, their approach uses hypothesis testing for determining uncertainty in policy evaluation for a pair of candidate policies based on sampling from model posteriors. This helps in ranking them and selection of better performing policy over the behavior policy in a safe manner. The work, however,

was limited to small flat state-spaces and did not explicitly evaluate uncertainty quantification. In contrast, our work produces a benchmark to primarily focus on uncertainty quantification of a system using offline data, rather than evaluating in terms of overall task performance.

In similar motivation, SOPR-T [31] also considers policy ranking from offline data and additional policy-value supervision. This is done by learning an encoded representation of a policy using a transformer based architecture and a scoring function over the representation. In order to learn the representation, they require a set of pre-defined policies, each labeled by its ground truth value with respect to an initial state distribution. Our framework does not assume the availability of such policy-value supervision and also puts an emphasis on uncertainty quantification, which is not evaluated by this work.

DOPE [17] studies OPE and devises a protocol that measures policy evaluation, ranking, and selection. For this purpose, the approach introduces a set of candidate policies along with their expected value over a distribution of initial states. Rather, in our work, we question the ability of a system to rank policies from any arbitrary state for a given horizon instead of limiting to initial state distribution only. This can help provide a more comprehensive view of uncertainty estimation across the state space.

**Confidence Intervals.** We make use of confidence intervals over policy value estimates for answering queries. [68] also studies confidence interval estimation over policy value estimates using trajectories generated by a different set of policies. Their approach uses importance sampling (IS) for unbiased value estimates, which suffers from high variance leading to loose confidence bounds. They also introduce the problem of *high confidence off-policy evaluation* and produce tighter bounds on estimates using improved concentration inequalities [43]. [45, 39, 44] further reduce variance in this problem by in-cooperating per-decision IS [55], power-mean [4], and self-normalization [27, 52]; respectively.

Another class of approaches is based on statistical bootstrapping [12]. [24] bootstraps learned MDP transition models in order to estimate lower confidence bounds on policy evaluation estimates with limited data. [33] suggests that confidence intervals of these bootstrapped estimates are not guaranteed to be accurate. In practice, they are shown to be overly confident especially for insufficient sample sizes and under-coverage of the data distribution. They suggest, that, in practice, this issue may be mitigated by inducing noisy rewards and regularization to learn smoother empirical transition and reward functions. Evaluating that claim within our OPCC framework is a potential direction for future work.

CoinDICE [9], and similar methods [66, 48, 75, 26] progressively focus on confidence intervals for OPE based on the formulation of certain optimization problems. These iterative optimization approaches [46, 47, 16, 15] for estimating policy value and confidence bounds induces a computational overload. This is undesirable in our framework which aims to rapidly answer queries of arbitrary horizons and policies making it computationally unsustainable. An interesting direction for future work is to consider generalizing this optimization-based approach to more flexibly handle arbitrary policies.

## 8  Summary

Properly quantifying uncertainty of complex models is a major open problem of practical significance in machine learning. Despite this fact, only a small fraction of the work in machine learning attempts to address this problem. Further, in areas such as offline RL, where methods for addressing uncertainty are developed, there is very little direct evaluation of uncertainty quantification. In recent years, there has been impressive progress on out-of-distribution detection for image classification, where quantifying uncertainty is a core problem. This has been largely driven by the availability of benchmarks that lower the overhead for conducting research and comparing methods. Currently, there is a lack of such benchmarks for sequential decision-making. The OPCC problem is a relatively simple problem to state, yet is rich enough to capture the essence of uncertainty quantification for sequential decision making. We hope that the OPCC benchmarks will inspire other researchers to develop new ideas for uncertainty quantification. Indeed, our pilot experiments show there is significant room to improve and that our understanding of current mechanisms is incomplete. Finally, we hope that this initial benchmark and baseline contribution is only the initial seed for the community at large to contribute to as progress is made.

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

# A  OPCC Benchmark Summary

As introduced in Section 4; we consider 7 environments for OPCC benchmark. These environments are relatively low-dimensional environments with non-image-based observations. This helps focus initial studies on fundamental OPCC capabilities, rather than simultaneously addressing the additional complexities that enter with lower-level perceptual observations such as images. In the following, we extend our discussion on these environments and elaborate PCQ construction steps. The outlined benchmark-construction schema is generic, which can be followed by others to extend the set of available OPCC benchmarks. Tables 4 and 5 and Figure 4 show a snapshot of OPCC benchmark components.

## A.1  Environments

**Maze2d (4 environments).** Figure 2 shows different maze environment configurations. Each environment has 4D observations giving the position and velocity of the ball being controlled and a 2D action space specifying the direction of movement. The goal in each environment is to control a rolling ball(green) to reach a goal(red) location. For our benchmarks, we used the *dense-reward version* of the environments. There are no terminal states in these environments and the episode ends after the maximum number of allowed time-steps reported in Table 4. The D4RL trajectory data sets were created by running a path-planning algorithm to navigate in the maze between different start and endpoints.

**Gym-Mujoco(3 environments).** Figure 3 shows considered gym-mujoco environments. These environments are qualitatively different from Maze2d in that they involve controlling periodic locomotion behavior based on continuous states and actions. Rather Maze2d is primarily about goal-based path planning (navigation) rather than controlling low-level locomotion.

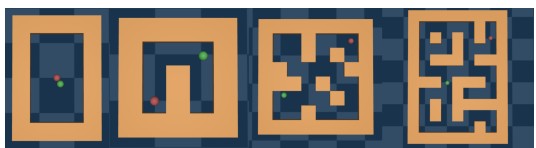

Figure 2: Maze2d tasks: open, umaze, medium, and large ( left to right).

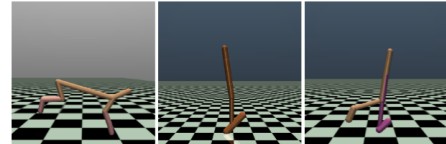

Figure 3: Gym-Mujoco tasks: half-cheetah, hopper, walker2d (left to right)

## A.2  Query Set Construction.

A possible starting point to create set of PCQs for OPCC is the off-policy evaluation (OPE) extension [17] to D4RL, which includes policies for a subset of the environments. In particular, one of the tasks considered is policy ranking, which is similar in spirit to OPCC. However, that extension of D4RL does not capture at least two important characteristics of OPCC. First, the evaluation protocols do not explicitly address measuring the quality of uncertainty quantification. Of course, this can be addressed by just extending the evaluation protocol and metrics.

Second, OPE ranking task from D4RL is currently limited to just ranking policies based on their expected values over the initial state distribution of each environment. In contrast, OPCC evaluations should involve sets of PCQs that cover a wide range of states that are both in-distribution and out-of-distribution relative to the offline data set. Further, it is desirable to select the PCQs in a way that spans some notion of PCQ difficulty. In particular, the notion of difficulty we consider here for a PCQ $(s, \pi, \hat{s}, \hat{\pi}, h)$ is directly related to the performance gap between the policies, i.e $\left|V^{\pi}(s, h) - V^{\hat{\pi}}(\hat{s}, h)\right|$. It is expected that all else being equal, larger gaps will reason in easier discrimination between policies. Indeed one of the initial challenges in developing the benchmarks was to try to create query sets that were not all too easy or too hard.

Based on the above considerations, we create the evaluation sets of PCQs for each environment via the following steps.

**Step 1: Policy Generation.** We first train[63]/formulate multiple policies for each environment to serve as the policies used for PCQ construction. For each environment, we have sufficiently distinct policies in terms of quality and behavior to support non-trivial PCQs. In each maze2d environment,

we formulated 5 policies for different goal locations. This helps in covering a wider state-action space of the grid. However, in our PCQs, they are evaluated over the actual task. In gym-mujoco, we trained 10 policies for different stride lengths and directions( forward and backward) of motion. Performance of these policies is shared in Table 5.

We chose to train new policies, rather than use policies from D4RL, to ensure they would be distinct from the behavior policies used to create the D4RL datasets. For each environment, we used the corresponding simulator and multiple runs of the PPO algorithm [63] to train a set of policies of varying quality.

**Step 2: Initial State Generation.** For each environment we generated a large set of potential initial states by running episodes of the random policy, the learned policies, and a mixture of random and learned policies. This produced a set of states that covered a wide range of the environment that extended well beyond the initial state distributions.

**Step 3: Candidate PCQ Generation.** For each horizon $h \in \{10, 20, 30, 40, 50\}$ we create a set of 1500 randomly constructed PCQs from the initial states and learned policies. This included explicitly creating random PCQs of the form $(s, \pi, s, \hat{\pi}, h)$ with $s$ a random initial state and $\pi, \hat{\pi}$ a random pair of the learned policies. In addition, we create a set of PCQs of the form $(s, \pi, \hat{s}, \hat{\pi}, h)$ in the same way, except that two random initial states are used instead of one.

**Step 4: PCQ Labeling and Selection.** For each generated PCQ from step 3 we used Monte-Carlo simulation via the environment simulator to accurately estimate $V^{\pi}(s, h)$ and $V^{\hat{\pi}}(\hat{s}, h)$ and removed any PCQ having a difference of less than 10 between the value of each side of a query. The motivation is to filter out PCQs that are the most ambiguous and more likely to act as a source of noise in evaluations. Finally, for each $h$, we randomly selected 1500 of the PCQs to include as the benchmark query set.

In Figure 4, we show scatter plots of $\left(V^{\pi}(s, h), V^{\hat{\pi}}(\hat{s}, h)\right)$ for the selected set of PCQs for each environment. Notice the lack of PCQs along the diagonal, which corresponds to the removal of ambiguous queries. Also note that the PCQs span a range of value gaps, which suggests that they span varying PCQs of varying difficult. If these plots showed a bias toward only large gap queries, then additional steps would be necessary to ensure that more variation was present in the selected query sets.

Table 4: Information about OPCC Benchmark comprising of environment details, datasets, and queries.

| ENV. | OBS. DIM. | ACTION DIM. | MAX. ENV. STEPS | DATASET TYPE | QUERY COUNT |
|---|---|---|---|---|---|
| MAZE2D-OPEN-V0 | | | 150 | | |
| MAZE2D-MEDIUM-V1 | 4 | 2 | 600 | 1M | 1500 |
| MAZE2D-UMAZE-V1 | | | 300 | | |
| MAZE2D-LARGE-V1 | | | 800 | | |
| HOPPER-V2 | 11 | 3 | | RANDOM, EXPERT, | |
| HALFCHEETAH-V2 | 17 | 6 | 1000 | MEDIUM, MEDIUM-REPLAY, | 1500 |
| WALKER2D-V2 | | | | MEDIUM-EXPERT | |

## A.3 Extending PCQs

As mentioned in Section 3.1, we can use PCQs for policy improvement as well. In particular, we can improve over policy $\pi$ at state $s$ by identifying an action $a'$ with higher action value than chosen by $\pi$. The corresponding PCQ for testing $a'$ is $(s, \pi, s, \pi', h)_M$, where $\pi'$ is the non-stationary policy that first takes action $a'$ and then follows $\pi$.

In practice, PCQs within an application domain need not be restricted to comparing policies via a single reward function. Rather there are often multiple quantities of interest to users. For example, a farm manager may be interested in understanding how two irrigation policies compare across multiple features of the future, such as cumulative water usage, plant stress, run off, etc. This can be facilitated by defining reward functions corresponding to each feature and issuing the appropriate PCQs.

Table 5: Performance of policies used in PCQs. These policies are trained using PPO [63] over the original environment task and hand-picked at different performance levels. We report mean( standard deviation) of policy performance over 20 episodes.

| ENV. | POLICY-1 | POLICY-2 | POLICY-3 | POLICY-4 | POLICY-5 |
|------|----------|----------|----------|----------|----------|
| MAZE2D-OPEN-V0 | 122(10) | 53(6) | 38(3) | 28(9) | 25(6) |
| MAZE2D-UMAZE-V1 | 189(97) | 59(59) | 84(26) | 33(2) | 20(2) |
| MAZE2D-MEDIUM-V1 | 344(251) | 41(28) | 42(57) | 41(53) | 17(35) |
| MAZE2D-LARGE-V1 | 436(304) | 38(37) | 31(95) | 3(7) | 1(3) |
| HALFCHEETAH-V2 | 6444(126) | 3117(11) | 1552(50) | 458(4) | −157(5) |
| HOPPER-V2 | 3299(276) | 2045(1) | 1508(2) | 1238(17) | 1007(4) |
| WALKER2D-V2 | 1927(12) | 1890(364) | 1840(267) | 1474(11) | 670(26) |

| ENV. | POLICY-6 | POLICY-7 | POLICY-8 | POLICY-9 | POLICY-10 |
|------|----------|----------|----------|----------|-----------|
| HALFCHEETAH-V2 | 1168(80) | 1044(112) | 785(303) | 94(40) | 4(8) |
| HOPPER-V2 | 860(45) | 851(4) | 582(4) | 194(1) | 82(5) |
| WALKER2D-V2 | 470(168) | 70(18) | −235(172) | −1330(160) | −1770(730) |

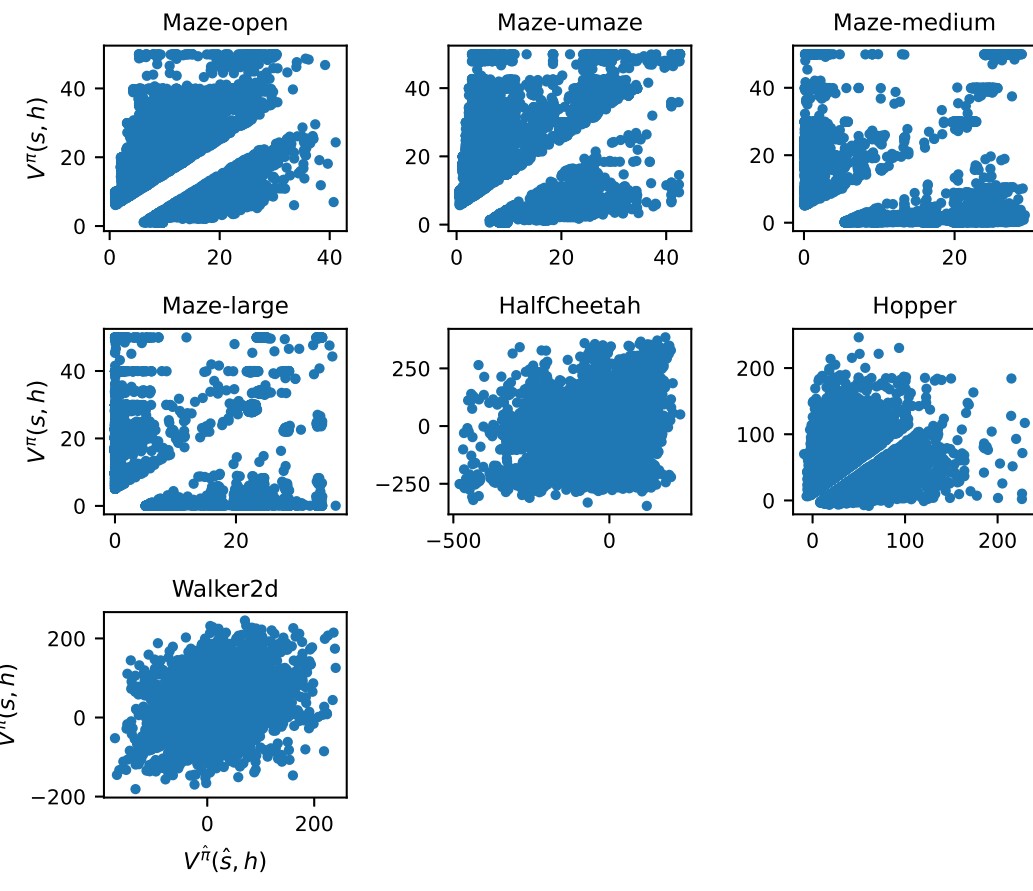

Figure 4: Scatter plot of the PCQ for each benchmark environment. For each PCQ $(s, \pi, \hat{s}, \hat{\pi}, h)$, we plot $V^\pi(s, h)$ vs. $V^{\hat{\pi}}(\hat{s}, h)$.

## B  OPCC Evaluation Metrics Intra-relation

A system gaining on AURCC indicates that it produces low risks at multiple coverage points. But, it doesn't necessarily help us understand the quality of the confidence produced. Similar AURCC could be achieved by another framework with a different set of coverage points. In order to understand this, we supplement our primary metric AURCC with $CR_K$ and RPP. A gain on $CR_K$ informs us about the diversity of coverage points produced by a system, in turn informing us about varying assigned confidences. This information supplements our AURCC information as a singleton view of $CR_K$ wouldn't tell us anything about risk. Also, a gain on RPP implies relatively low confidences were assigned to queries as compared to correctly answered queries. This could be achieved by a binary confidence indicator as well, hiding information about exhibited confidence diversity. Thereby, a combination of all three metrics gives us a better understanding of the uncertainty estimation.

## C  Experiment Implementation Detail

Two additional details are important to note for our experiments. First, as is customary in model-based RL (including ORL), we are using a pre-defined episode termination function rather than a learned one. We have found that this can significantly impact the performance of model-based RL systems and also our OPCC evaluations. Second, we clipped predicted observations and rewards to keep them within the bounds of the available data sets, which is also a common practice in ORL that we found to be important.

## D  Dynamics Model Ablations

In this section, we extend our discussion on the impact of considered dynamics architecture choices on OPCC benchmark.

**Impact of Ensemble Size.** We consider the impact of ensemble size for our baselines. Table 6 shows the results for ensemble sizes ranging from 10 to 100. Our prior expectation was that performance would increase with significant increase in ensemble-size. In general, we do not see statistically significant differences between ensemble sized for AURCC based on our current experimental budget (i.e. confidence intervals intersect). However, based on trends in the means, there is weak evidence of improved AURCC. The exception is HalfCheetah, where for AURCC, the trends is opposite of the expectation. However, the differences in means tends to be small, suggesting that ensemble size is not having a large impact even if more computational budget were devoted to support statistical significance.

For RPP and Coverage Resolution ($CR_k$) there is typically a statistically significant improvement from ensemble size 10 to 100. The exceptions are umaze and medium-maze where losses are very small for all ensemble sizes. Overall, however, differences are relatively small in magnitude. This may be due to the ensembles not being diverse enough, or the base models used to construct the ensembles are not accurate enough. These results demonstrate the value of the OPCC benchmarks in being able to explicitly test hypotheses about uncertainty quantification, rather than relying on downstream results that may be impacted by many possible factors.

**Randomized Constant Priors**. In order to encourage diversity, we introduce *randomized constant priors* in our ensemble models. These are suggested to encourage extrapolation diversity, especially on out-of-distribution state-action pairs, which could improve disagreement-based uncertainty estimates. However, when we included the constant priors in our model, we didn't find significant improvements in our evaluation metrics as shown in Table 7 and Figure 6. We use the same architecture as the dynamics model for *prior* with random weights and scale them with *"prior-scale"* before adding them to ensemble models. A prior-scale of 0 indicates no usage of prior. In the case of maze2d, we generally observe a slight (but statistically insignificant) reduction in *AURCC*, whereas *RPP* and $CR_K$ tends to remain same. Large maze environment has a significant decrease in AURCC with randomized constant prior. On the contrary, in the case of gym-mujoco, we generally observe a slight (but statistically insignificant) increase in *AURCC*, *RPP*, and $CR_K$.

Prior work by **?** demonstrated improvement in end-task RL performance by having an ensemble of DQN (**?**) models with randomized constant priors. However, explicit analysis of the uncertainty quantification was not provided. Our observations suggests that randomized constant priors do not

Table 6: Evaluation metrics for *ensemble-size* comparison in *gym-mujoco* and *maze2d* environments. We train 5 (seed) ensemble dynamics of size 100 for each dataset and start with an ensemble of 10 models for OPCC metrics estimation. Thereafter, we incrementally increase their count and determine impact on metrics mean and confidence intervals (95%).

| ENV. | ENSEMBLE COUNT | AURCC($\downarrow$) | RPP($\downarrow$) | CR$_{10}$($\uparrow$) | LOSS($\downarrow$) |
|---|---|---|---|---|---|
| HOPPER | 10 | $0.141 \pm 0.005$ | $0.02 \pm 0.004$ | $0.316 \pm 0.029$ | $0.273 \pm 0.005$ |
| | 20 | $0.141 \pm 0.005$ | $0.024 \pm 0.004$ | $0.344 \pm 0.037$ | $0.272 \pm 0.004$ |
| | 40 | $0.141 \pm 0.005$ | $0.027 \pm 0.004$ | $0.388 \pm 0.04$ | $0.27 \pm 0.004$ |
| | 80 | $0.141 \pm 0.005$ | $0.03 \pm 0.004$ | $0.416 \pm 0.038$ | $0.269 \pm 0.004$ |
| | 100 | $0.141 \pm 0.005$ | $0.031 \pm 0.005$ | $0.428 \pm 0.034$ | $0.268 \pm 0.004$ |
| HALF CHEETAH | 10 | $0.209 \pm 0.004$ | $0.028 \pm 0.004$ | $0.32 \pm 0.029$ | $0.378 \pm 0.004$ |
| | 20 | $0.213 \pm 0.004$ | $0.034 \pm 0.004$ | $0.36 \pm 0.04$ | $0.376 \pm 0.004$ |
| | 40 | $0.215 \pm 0.004$ | $0.039 \pm 0.005$ | $0.4 \pm 0.035$ | $0.374 \pm 0.004$ |
| | 80 | $0.219 \pm 0.005$ | $0.043 \pm 0.005$ | $0.452 \pm 0.04$ | $0.374 \pm 0.004$ |
| | 100 | $0.219 \pm 0.005$ | $0.044 \pm 0.005$ | $0.46 \pm 0.04$ | $0.374 \pm 0.004$ |
| WALKER 2D | 10 | $0.072 \pm 0.001$ | $0.007 \pm 0.002$ | $0.256 \pm 0.03$ | $0.16 \pm 0.002$ |
| | 20 | $0.071 \pm 0.001$ | $0.008 \pm 0.002$ | $0.268 \pm 0.038$ | $0.16 \pm 0.002$ |
| | 40 | $0.07 \pm 0.001$ | $0.01 \pm 0.003$ | $0.28 \pm 0.046$ | $0.16 \pm 0.002$ |
| | 80 | $0.068 \pm 0.001$ | $0.01 \pm 0.003$ | $0.284 \pm 0.049$ | $0.159 \pm 0.002$ |
| | 100 | $0.068 \pm 0.001$ | $0.011 \pm 0.003$ | $0.3 \pm 0.051$ | $0.159 \pm 0.002$ |
| OPEN | 10 | $0.033 \pm 0.004$ | $0.009 \pm (< 0.001)$ | $0.48 \pm 0.035$ | $0.127 \pm 0.015$ |
| | 20 | $0.031 \pm 0.003$ | $0.01 \pm (< 0.001)$ | $0.5 \pm (< 0.001)$ | $0.117 \pm 0.019$ |
| | 40 | $0.03 \pm 0.003$ | $0.011 \pm 0.001$ | $0.5 \pm (< 0.001)$ | $0.11 \pm 0.014$ |
| | 80 | $0.029 \pm 0.002$ | $0.012 \pm 0.001$ | $0.5 \pm (< 0.001)$ | $0.108 \pm 0.007$ |
| | 100 | $0.029 \pm 0.001$ | $0.012 \pm 0.001$ | $0.5 \pm (< 0.001)$ | $0.107 \pm 0.005$ |
| UMAZE | 10 | $0.011 \pm 0.002$ | $0.002 \pm (< 0.001)$ | $0.3 \pm (< 0.001)$ | $0.073 \pm 0.006$ |
| | 20 | $0.009 \pm 0.001$ | $0.002 \pm (< 0.001)$ | $0.3 \pm (< 0.001)$ | $0.074 \pm 0.004$ |
| | 40 | $0.008 \pm 0.001$ | $0.002 \pm (< 0.001)$ | $0.3 \pm (< 0.001)$ | $0.077 \pm 0.004$ |
| | 80 | $0.008 \pm 0.001$ | $0.002 \pm (< 0.001)$ | $0.3 \pm (< 0.001)$ | $0.075 \pm 0.004$ |
| | 100 | $0.008 \pm 0.001$ | $0.002 \pm (< 0.001)$ | $0.3 \pm (< 0.001)$ | $0.075 \pm 0.003$ |
| MEDIUM | 10 | $0.001 \pm (< 0.001)$ | $0.0 \pm (< 0.001)$ | $0.2 \pm (< 0.001)$ | $0.023 \pm 0.007$ |
| | 20 | $0.001 \pm (< 0.001)$ | $0.0 \pm (< 0.001)$ | $0.2 \pm (< 0.001)$ | $0.024 \pm 0.006$ |
| | 40 | $0.001 \pm (< 0.001)$ | $0.0 \pm (< 0.001)$ | $0.2 \pm (< 0.001)$ | $0.021 \pm 0.003$ |
| | 80 | $0.001 \pm (< 0.001)$ | $0.0 \pm (< 0.001)$ | $0.2 \pm (< 0.001)$ | $0.022 \pm 0.001$ |
| | 100 | $0.001 \pm (< 0.001)$ | $0.0 \pm (< 0.001)$ | $0.2 \pm (< 0.001)$ | $0.022 \pm 0.001$ |
| LARGE | 10 | $0.168 \pm 0.044$ | $0.051 \pm 0.011$ | $0.6 \pm (< 0.001)$ | $0.307 \pm 0.061$ |
| | 20 | $0.149 \pm 0.039$ | $0.057 \pm 0.015$ | $0.78 \pm 0.066$ | $0.269 \pm 0.065$ |
| | 40 | $0.138 \pm 0.031$ | $0.056 \pm 0.011$ | $0.82 \pm 0.035$ | $0.264 \pm 0.044$ |
| | 80 | $0.146 \pm 0.019$ | $0.061 \pm 0.007$ | $0.82 \pm 0.035$ | $0.269 \pm 0.027$ |
| | 100 | $0.14 \pm 0.015$ | $0.062 \pm 0.004$ | $0.82 \pm 0.035$ | $0.251 \pm 0.029$ |

appear to improve uncertainty quantification at least as measured through our OPCC benchmarks. Further investigation is necessary to better understand the performance differences observed in **?**. An interesting direction of future work is to consider other previously proposed mechanisms for improving ensemble diversity within the OPCC framework.

**Dynamics Model Types.** We ablated our base model's feed-forward(FF) architecture with Autoregressive(AR) architecture suggested by **?**. In Table 8 and Figure 7, we do not observe significant evidence in favor of the AR model with respect to OPCC performance. There is a marginal, but no statistical significant reduction in *AURCC* in some cases. We do see an increase in coverage resolution ($CR_k$) for the gym-mujoco environments when using the AR model, while it remains the same for the maze2d environments. This may be due to the additional uncertainty propagation that can occur during auto-regressive inference of each dimension, especially in the higher-dimensional gym-mujoco environments. Currently our results do not suggest that the extra computational cost of the AR model compared to FF is worthwhile with respect to uncertainty quantification as measured via OPCC. This

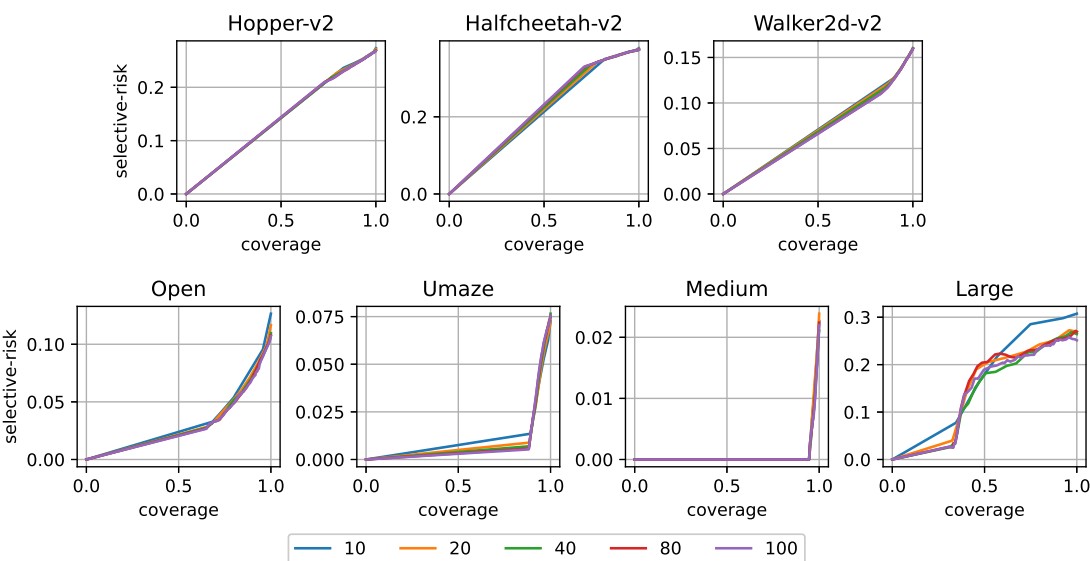

Figure 5: Selective-risk coverage curves for *ensemble-count* in *gym-mujoco* (top-row) and maze2d (bottom-row) environments

Table 7: Evaluation metrics for *prior-scale* comparison in *gym-mujoco* environments comprising of mean and confidence interval(95%) over 50 samples belonging to 5 (seed) dynamics models for each of the 5 datasets. Prior scale of 0 means no randomized constant prior is added.

| ENV. | PRIOR SCALE | AURCC($\downarrow$) | RPP($\downarrow$) | CR$_{10}$($\uparrow$) | LOSS($\downarrow$) |
|---|---|---|---|---|---|
| HOPPER | 0 | $0.141 \pm 0.005$ | $0.031 \pm 0.005$ | $0.428 \pm 0.034$ | $0.268 \pm 0.004$ |
|  | 5 | $0.145 \pm 0.003$ | $0.045 \pm 0.004$ | $0.616 \pm 0.05$ | $0.269 \pm 0.004$ |
| HALF CHEETAH | 0 | $0.219 \pm 0.005$ | $0.044 \pm 0.005$ | $0.46 \pm 0.04$ | $0.374 \pm 0.004$ |
|  | 5 | $0.236 \pm 0.005$ | $0.067 \pm 0.004$ | $0.768 \pm 0.081$ | $0.373 \pm 0.006$ |
| WALKER 2D | 0 | $0.068 \pm 0.001$ | $0.011 \pm 0.003$ | $0.3 \pm 0.051$ | $0.159 \pm 0.002$ |
|  | 5 | $0.057 \pm 0.001$ | $0.017 \pm 0.002$ | $0.44 \pm 0.04$ | $0.159 \pm 0.002$ |
| OPEN | 0 | $0.029 \pm 0.001$ | $0.012 \pm 0.001$ | $0.5 \pm (< 0.001)$ | $0.107 \pm 0.005$ |
|  | 5 | $0.032 \pm 0.001$ | $0.012 \pm (< 0.001)$ | $0.5 \pm (< 0.001)$ | $0.115 \pm 0.008$ |
| UMAZE | 0 | $0.008 \pm 0.001$ | $0.002 \pm (< 0.001)$ | $0.3 \pm (< 0.001)$ | $0.075 \pm 0.003$ |
|  | 5 | $0.006 \pm 0.001$ | $0.002 \pm (< 0.001)$ | $0.3 \pm (< 0.001)$ | $0.071 \pm 0.002$ |
| MEDIUM | 0 | $0.001 \pm (< 0.001)$ | $0.0 \pm (< 0.001)$ | $0.2 \pm (< 0.001)$ | $0.022 \pm 0.001$ |
|  | 5 | $0.0 \pm (< 0.001)$ | $0.0 \pm (< 0.001)$ | $0.2 \pm (< 0.001)$ | $0.02 \pm 0.003$ |
| LARGE | 0 | $0.14 \pm 0.015$ | $0.062 \pm 0.004$ | $0.82 \pm 0.035$ | $0.251 \pm 0.029$ |
|  | 5 | $0.104 \pm 0.017$ | $0.051 \pm 0.012$ | $0.82 \pm 0.035$ | $0.197 \pm 0.017$ |

may be due to the environments not needed to represent multi-modal output distribution, which is where the AR model could have a distinct advantage.

**Determinism.** Our baseline model is a deterministic version of the stochastic model defined in **?**, trained via regression loss. A classic improvement is to induce stochasticity into the model by learning a normal distribution over the next observation rather than a point estimate. We experimented with this modification and provide results in Table 9 and Figure 8). Though, limitations of deterministic models are well-understood for stochastic environments, it turns out we don't gain significantly with stochastic models in our pilot run. This is possibly due to deterministic nature of maze environments and low stochasticity in gym-mujoco case.

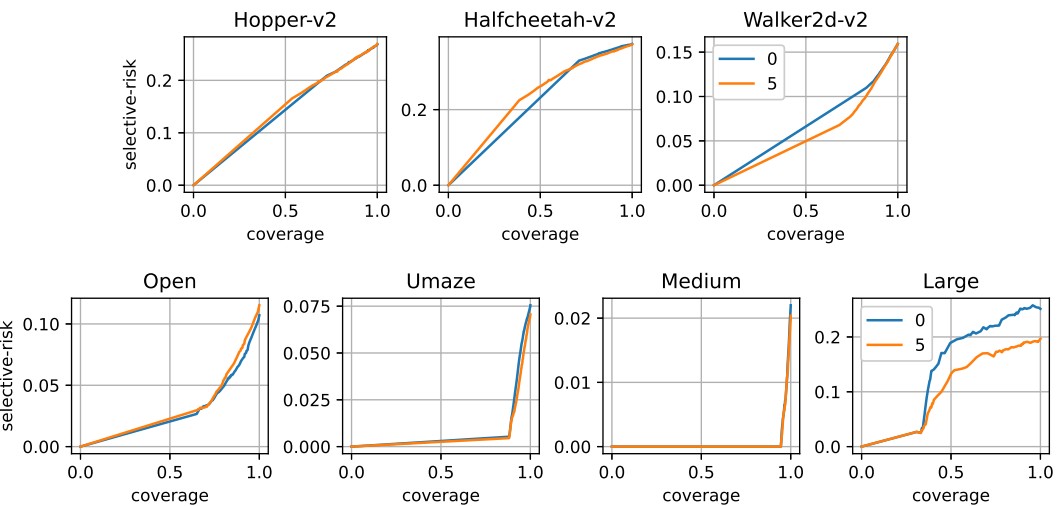

Figure 6: Selective-risk coverage curves for *prior-scale* in *gym-mujoco* and *maze* environments

Table 8: Evaluation metrics for *dynamics-type* comparison in *gym-mujoco* environments comprising of mean and confidence interval(95%) estimates over 50 samples belonging to 5 (seeds) dynamics models for each of the 5 datasets. Here, 'AR' and 'FF' implies auto-regressive model and feed-forward model, respectively.

| ENV. | DYNAMICS TYPE | AURCC($\downarrow$) | RPP($\downarrow$) | CR$_{10}$($\uparrow$) | LOSS($\downarrow$) |
|---|---|---|---|---|---|
| HOPPER | AR | $0.139 \pm 0.007$ | $0.034 \pm 0.005$ | $0.48 \pm 0.042$ | $0.268 \pm 0.007$ |
| | FF | $0.141 \pm 0.005$ | $0.031 \pm 0.005$ | $0.428 \pm 0.034$ | $0.268 \pm 0.004$ |
| HALF CHEETAH | AR | $0.249 \pm 0.008$ | $0.068 \pm 0.006$ | $0.736 \pm 0.087$ | $0.379 \pm 0.003$ |
| | FF | $0.219 \pm 0.005$ | $0.044 \pm 0.005$ | $0.46 \pm 0.04$ | $0.374 \pm 0.004$ |
| WALKER 2D | AR | $0.065 \pm 0.002$ | $0.013 \pm 0.002$ | $0.356 \pm 0.046$ | $0.158 \pm 0.002$ |
| | FF | $0.068 \pm 0.001$ | $0.011 \pm 0.003$ | $0.3 \pm 0.051$ | $0.159 \pm 0.002$ |
| OPEN | AR | $0.034 \pm 0.003$ | $0.014 \pm 0.002$ | $0.5 \pm (< 0.001)$ | $0.123 \pm 0.006$ |
| | FF | $0.029 \pm 0.001$ | $0.012 \pm 0.001$ | $0.5 \pm (< 0.001)$ | $0.107 \pm 0.005$ |
| UMAZE | AR | $0.007 \pm 0.001$ | $0.002 \pm (< 0.001)$ | $0.3 \pm (< 0.001)$ | $0.07 \pm 0.002$ |
| | FF | $0.008 \pm 0.001$ | $0.002 \pm (< 0.001)$ | $0.3 \pm (< 0.001)$ | $0.075 \pm 0.003$ |
| MEDIUM | AR | $0.001 \pm (< 0.001)$ | $0.0 \pm (< 0.001)$ | $0.2 \pm (< 0.001)$ | $0.031 \pm 0.001$ |
| | FF | $0.001 \pm (< 0.001)$ | $0.0 \pm (< 0.001)$ | $0.2 \pm (< 0.001)$ | $0.022 \pm 0.001$ |
| LARGE | AR | $0.131 \pm 0.017$ | $0.06 \pm 0.003$ | $0.8 \pm (< 0.001)$ | $0.233 \pm 0.044$ |
| | FF | $0.14 \pm 0.015$ | $0.062 \pm 0.004$ | $0.82 \pm 0.035$ | $0.251 \pm 0.029$ |

**Normalization of input state-space.** In Table 10 and Figure 9, we investigate the impact of learning dynamics with normalized state-space. Here, *'True'* implies the dynamics was learned with normalized state-space and *'False'* implies otherwise. There is a marginal performance difference between either choice for MuJoco environments. Maze2D environments show a mix of results with normalization benefiting Umaze and hurting Large-Maze. Performance of Medium-Maze and Open-Maze is not impacted significantly.

# E    Extended Experimental Data

In the following sub-sections, we share OPCC metrics for various ablations discussed in our primary experiments(Section 6) regarding dataset quality, uncertainty types, and query horizons. In each

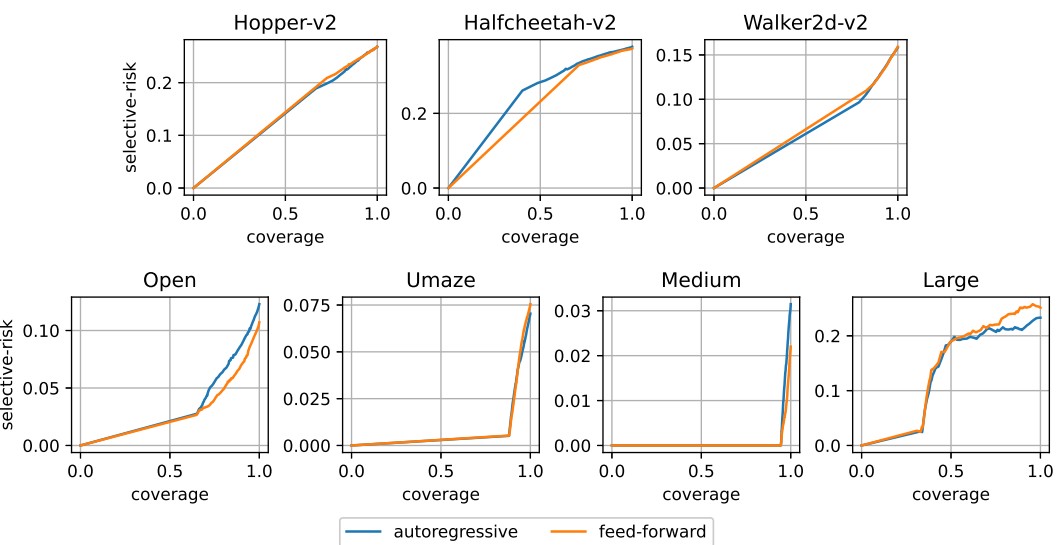

Figure 7: Selective-risk coverage curves for *dynamics-type* in *gym-mujoco* and *maze* environments

Table 9: Evaluation metrics for *deterministic* model comparison in *gym-mujoco* and *maze2d* environments. 'True' implies a deterministic model, whereas 'False' implies a stochastic model.

| ENV. | DETER-MINISTIC | AURCC($\downarrow$) | RPP($\downarrow$) | CR$_{10}$($\uparrow$) | LOSS($\downarrow$) |
|---|---|---|---|---|---|
| HOPPER | FALSE | $0.138 \pm 0.002$ | $0.039 \pm 0.002$ | $0.524 \pm 0.039$ | $0.26 \pm 0.003$ |
|  | TRUE | $0.141 \pm 0.005$ | $0.031 \pm 0.005$ | $0.428 \pm 0.034$ | $0.268 \pm 0.004$ |
| HALF CHEETAH | FALSE | $0.229 \pm 0.007$ | $0.054 \pm 0.006$ | $0.568 \pm 0.057$ | $0.377 \pm 0.005$ |
|  | TRUE | $0.219 \pm 0.005$ | $0.044 \pm 0.005$ | $0.46 \pm 0.04$ | $0.374 \pm 0.004$ |
| WALKER 2D | FALSE | $0.064 \pm (< 0.001)$ | $0.012 \pm 0.001$ | $0.328 \pm 0.024$ | $0.16 \pm 0.002$ |
|  | TRUE | $0.068 \pm 0.001$ | $0.011 \pm 0.003$ | $0.3 \pm 0.051$ | $0.159 \pm 0.002$ |
| OPEN | FALSE | $0.037 \pm 0.001$ | $0.01 \pm (< 0.001)$ | $0.4 \pm (< 0.001)$ | $0.143 \pm 0.005$ |
|  | TRUE | $0.029 \pm 0.001$ | $0.012 \pm 0.001$ | $0.5 \pm (< 0.001)$ | $0.107 \pm 0.005$ |
| UMAZE | FALSE | $0.004 \pm (< 0.001)$ | $0.001 \pm (< 0.001)$ | $0.2 \pm (< 0.001)$ | $0.059 \pm 0.004$ |
|  | TRUE | $0.008 \pm 0.001$ | $0.002 \pm (< 0.001)$ | $0.3 \pm (< 0.001)$ | $0.075 \pm 0.003$ |
| MEDIUM | FALSE | $0.0 \pm (< 0.001)$ | $0.0 \pm (< 0.001)$ | $0.2 \pm (< 0.001)$ | $0.003 \pm (< 0.001)$ |
|  | TRUE | $0.001 \pm (< 0.001)$ | $0.0 \pm (< 0.001)$ | $0.2 \pm (< 0.001)$ | $0.022 \pm 0.001$ |
| LARGE | FALSE | $0.152 \pm 0.01$ | $0.058 \pm 0.007$ | $0.54 \pm 0.043$ | $0.167 \pm 0.003$ |
|  | TRUE | $0.14 \pm 0.015$ | $0.062 \pm 0.004$ | $0.82 \pm 0.035$ | $0.251 \pm 0.029$ |

table-cell, we show mean and confidence interval at *95% confidence level* for corresponding metrics, estimated by evaluating 5 dynamics runs over each dataset of the corresponding environment.

### E.1 Dataset Quality

### E.2 Uncertainty Types

In the following, *'EV, PCI, U-PCI'* refer to Ensemble-Voting, Paired Confidence interval, and Unpaired Confidence Interval, respectively.

### E.3 Horizon

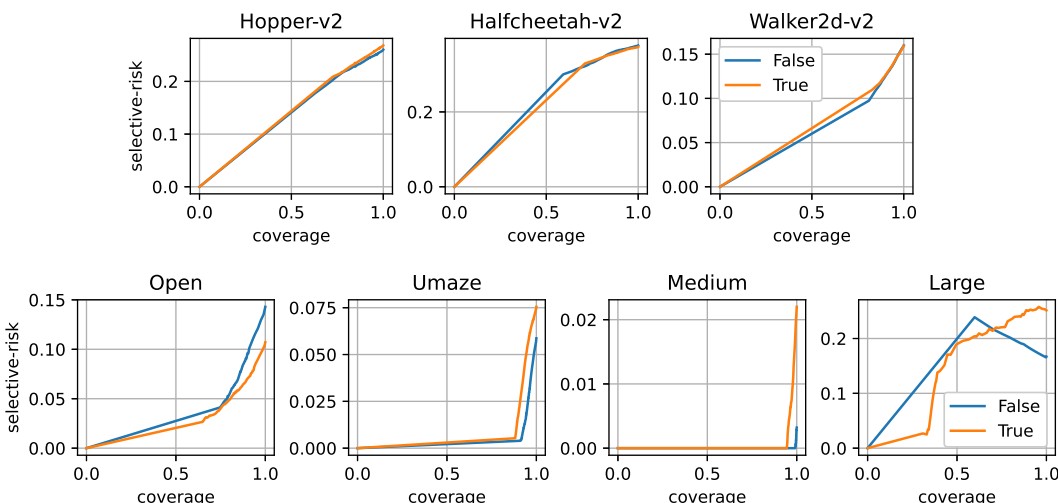

Figure 8: Selective-risk coverage curves for *deterministic* choice for dynamics model in *maze2d* and *gym-mujoco* environments. 'True' implies a deterministic model, whereas 'False' implies a stochastic model.

Table 10: Evaluation metrics for *normalize* comparison in *gym-mujoco* environments

| ENV. | NORMALIZE | AURCC($\downarrow$) | RPP($\downarrow$) | CR$_{10}$($\uparrow$) | LOSS($\downarrow$) |
|---|---|---|---|---|---|
| HOPPER | FALSE | $0.128 \pm 0.002$ | $0.017 \pm 0.002$ | $0.3 \pm 0.025$ | $0.255 \pm 0.003$ |
| | TRUE | $0.141 \pm 0.005$ | $0.031 \pm 0.005$ | $0.428 \pm 0.034$ | $0.268 \pm 0.004$ |
| HALF CHEETAH | FALSE | $0.228 \pm 0.006$ | $0.053 \pm 0.007$ | $0.548 \pm 0.07$ | $0.376 \pm 0.003$ |
| | TRUE | $0.219 \pm 0.005$ | $0.044 \pm 0.005$ | $0.46 \pm 0.04$ | $0.374 \pm 0.004$ |
| WALKER 2D | FALSE | $0.078 \pm 0.007$ | $0.013 \pm 0.005$ | $0.316 \pm 0.073$ | $0.17 \pm 0.009$ |
| | TRUE | $0.068 \pm 0.001$ | $0.011 \pm 0.003$ | $0.3 \pm 0.051$ | $0.159 \pm 0.002$ |
| OPEN | FALSE | $0.033 \pm 0.001$ | $0.014 \pm (< 0.001)$ | $0.5 \pm (< 0.001)$ | $0.108 \pm 0.004$ |
| | TRUE | $0.029 \pm 0.001$ | $0.012 \pm 0.001$ | $0.5 \pm (< 0.001)$ | $0.107 \pm 0.005$ |
| UMAZE | FALSE | $0.003 \pm (< 0.001)$ | $0.002 \pm (< 0.001)$ | $0.3 \pm (< 0.001)$ | $0.047 \pm 0.004$ |
| | TRUE | $0.008 \pm 0.001$ | $0.002 \pm (< 0.001)$ | $0.3 \pm (< 0.001)$ | $0.075 \pm 0.003$ |
| MEDIUM | FALSE | $0.0 \pm (< 0.001)$ | $0.0 \pm (< 0.001)$ | $0.2 \pm (< 0.001)$ | $0.017 \pm 0.002$ |
| | TRUE | $0.001 \pm (< 0.001)$ | $0.0 \pm (< 0.001)$ | $0.2 \pm (< 0.001)$ | $0.022 \pm 0.001$ |
| LARGE | FALSE | $0.215 \pm 0.027$ | $0.067 \pm 0.007$ | $0.82 \pm 0.035$ | $0.402 \pm 0.037$ |
| | TRUE | $0.14 \pm 0.015$ | $0.062 \pm 0.004$ | $0.82 \pm 0.035$ | $0.251 \pm 0.029$ |

Table 11: Evaluation metrics for *dataset-types* comparison in *maze* environments. This includes mean and confidence intervals estimates at $95\%$ confidence level for metrics corresponding to 5 (seed) dynamics trained over each dataset.

| ENV. | DATASET QUALITY | AURCC($\downarrow$) | RPP($\downarrow$) | CR$_{10}$($\uparrow$) | LOSS($\downarrow$) |
|---|---|---|---|---|---|
| OPEN | 1M | $0.029 \pm 0.001$ | $0.012 \pm 0.001$ | $0.5 \pm (< 0.001)$ | $0.107 \pm 0.005$ |
| UMAZE | 1M | $0.008 \pm 0.001$ | $0.002 \pm (< 0.001)$ | $0.3 \pm (< 0.001)$ | $0.075 \pm 0.003$ |
| MEDIUM | 1M | $0.001 \pm (< 0.001)$ | $0.0 \pm (< 0.001)$ | $0.2 \pm (< 0.001)$ | $0.022 \pm 0.001$ |
| LARGE | 1M | $0.14 \pm 0.015$ | $0.062 \pm 0.004$ | $0.82 \pm 0.035$ | $0.251 \pm 0.029$ |

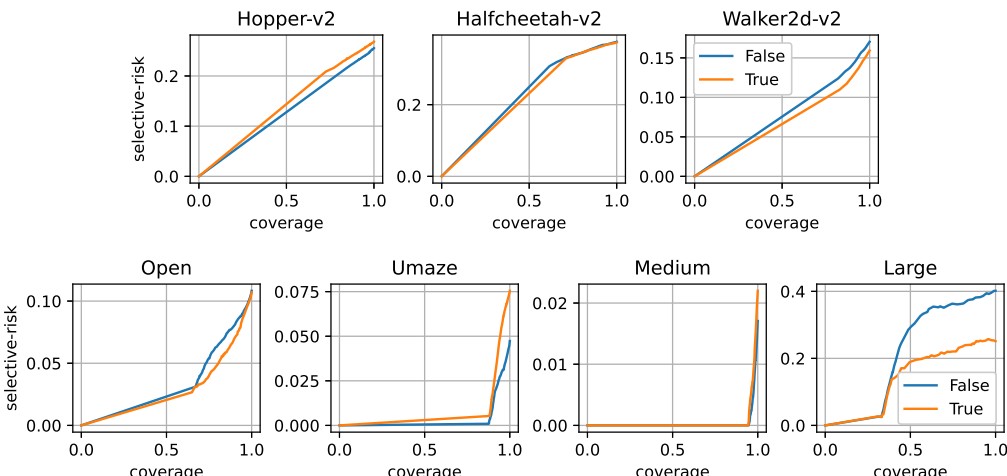

Figure 9: Selective-risk coverage curves for *input state-spce normalization* ablation in *maze* and *gym-mujoco* environments. 'True' implies the state-space is normalized and 'False' implies otherwise.

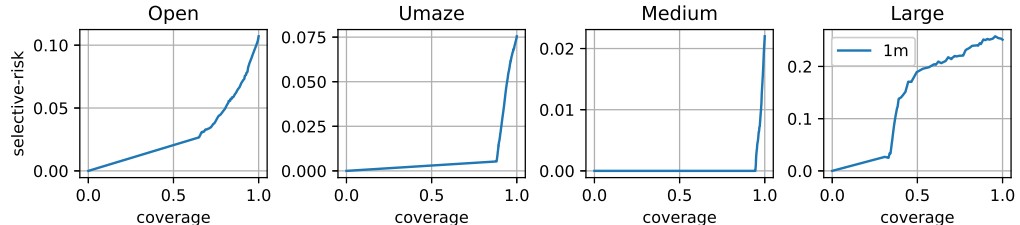

Figure 10: Selective-risk coverage curve for '1m' dataset in *maze* environments. This is the complete navigation dataset of *1 million* transactions.

Table 12: Evaluation metrics for *uncertainty-type* comparison in *maze* environments

| ENV. | UNCERTAINTY TYPE | AURCC($\downarrow$) | RPP($\downarrow$) | CR$_{10}$($\uparrow$) | LOSS($\downarrow$) |
|---|---|---|---|---|---|
| OPEN | EV | $0.029 \pm 0.001$ | $0.012 \pm 0.001$ | $0.5 \pm (< 0.001)$ | $0.107 \pm 0.005$ |
| | PCI | $0.057 \pm 0.004$ | $0.012 \pm 0.001$ | $0.38 \pm 0.035$ | $0.168 \pm 0.008$ |
| | U-PCI | $0.05 \pm 0.002$ | $0.012 \pm 0.001$ | $0.4 \pm (< 0.001)$ | $0.168 \pm 0.008$ |
| UMAZE | EV | $0.008 \pm 0.001$ | $0.002 \pm (< 0.001)$ | $0.3 \pm (< 0.001)$ | $0.075 \pm 0.003$ |
| | PCI | $0.035 \pm 0.002$ | $0.001 \pm (< 0.001)$ | $0.2 \pm (< 0.001)$ | $0.084 \pm 0.003$ |
| | U-PCI | $0.017 \pm 0.002$ | $0.001 \pm (< 0.001)$ | $0.2 \pm (< 0.001)$ | $0.084 \pm 0.003$ |
| MEDIUM | EV | $0.001 \pm (< 0.001)$ | $0.0 \pm (< 0.001)$ | $0.2 \pm (< 0.001)$ | $0.022 \pm 0.001$ |
| | PCI | $0.001 \pm (< 0.001)$ | $0.0 \pm (< 0.001)$ | $0.2 \pm (< 0.001)$ | $0.009 \pm 0.001$ |
| | U-PCI | $0.001 \pm (< 0.001)$ | $0.0 \pm (< 0.001)$ | $0.2 \pm (< 0.001)$ | $0.009 \pm 0.001$ |
| LARGE | EV | $0.14 \pm 0.015$ | $0.062 \pm 0.004$ | $0.82 \pm 0.035$ | $0.251 \pm 0.029$ |
| | PCI | $0.139 \pm 0.021$ | $0.037 \pm 0.018$ | $0.42 \pm 0.086$ | $0.218 \pm 0.028$ |
| | U-PCI | $0.155 \pm 0.014$ | $0.048 \pm 0.006$ | $0.46 \pm 0.043$ | $0.218 \pm 0.028$ |

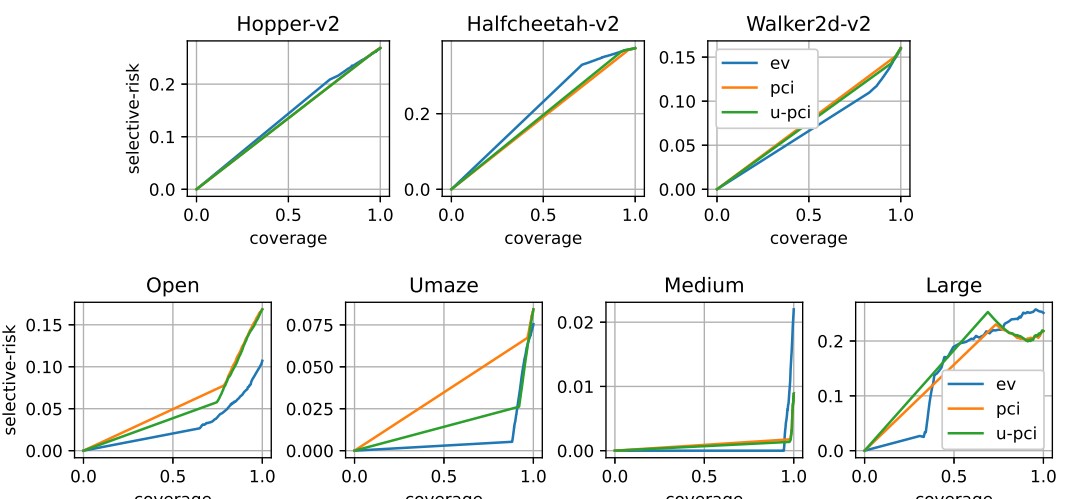

Figure 11: Selective-risk coverage curves for *uncertainty-type* in *gym-mujoco* and *maze2d* environments.

Table 13: Evaluation metrics for *horizon* comparison in *maze* environments

| ENV. | HORIZON | AURCC($\downarrow$) | RPP($\downarrow$) | CR$_{10}$($\uparrow$) | LOSS($\downarrow$) |
|---|---|---|---|---|---|
| OPEN | 20.0 | $0.005 \pm (< 0.001)$ | $0.001 \pm (< 0.001)$ | $0.2 \pm (< 0.001)$ | $0.029 \pm 0.001$ |
| | 30.0 | $0.015 \pm 0.002$ | $0.007 \pm 0.001$ | $0.5 \pm (< 0.001)$ | $0.104 \pm 0.006$ |
| | 40.0 | $0.036 \pm 0.002$ | $0.016 \pm 0.001$ | $0.6 \pm (< 0.001)$ | $0.119 \pm 0.007$ |
| | 50.0 | $0.062 \pm 0.002$ | $0.025 \pm 0.001$ | $0.6 \pm (< 0.001)$ | $0.148 \pm 0.006$ |
| UMAZE | 20.0 | $0.0 \pm (< 0.001)$ | $0.0 \pm (< 0.001)$ | $0.2 \pm (< 0.001)$ | $0.007 \pm 0.001$ |
| | 30.0 | $0.0 \pm (< 0.001)$ | $0.0 \pm (< 0.001)$ | $0.2 \pm (< 0.001)$ | $0.012 \pm 0.002$ |
| | 40.0 | $0.006 \pm 0.001$ | $0.002 \pm (< 0.001)$ | $0.2 \pm (< 0.001)$ | $0.048 \pm 0.003$ |
| | 50.0 | $0.035 \pm 0.002$ | $0.008 \pm 0.001$ | $0.4 \pm (< 0.001)$ | $0.195 \pm 0.007$ |
| MEDIUM | 20.0 | $0.0 \pm (< 0.001)$ | $0.0 \pm (< 0.001)$ | $0.2 \pm (< 0.001)$ | $0.006 \pm 0.001$ |
| | 30.0 | $0.0 \pm (< 0.001)$ | $0.0 \pm (< 0.001)$ | $0.2 \pm (< 0.001)$ | $0.019 \pm 0.001$ |
| | 40.0 | $0.001 \pm (< 0.001)$ | $0.0 \pm (< 0.001)$ | $0.2 \pm (< 0.001)$ | $0.032 \pm 0.003$ |
| | 50.0 | $0.001 \pm (< 0.001)$ | $0.001 \pm (< 0.001)$ | $0.2 \pm (< 0.001)$ | $0.031 \pm (< 0.001)$ |
| LARGE | 20.0 | $0.028 \pm (< 0.001)$ | $0.021 \pm (< 0.001)$ | $0.62 \pm 0.035$ | $0.059 \pm (< 0.001)$ |
| | 30.0 | $0.118 \pm 0.013$ | $0.047 \pm 0.004$ | $0.8 \pm (< 0.001)$ | $0.295 \pm 0.037$ |
| | 40.0 | $0.218 \pm 0.018$ | $0.087 \pm 0.004$ | $0.82 \pm 0.035$ | $0.321 \pm 0.027$ |
| | 50.0 | $0.16 \pm 0.018$ | $0.07 \pm 0.005$ | $0.92 \pm 0.035$ | $0.247 \pm 0.038$ |

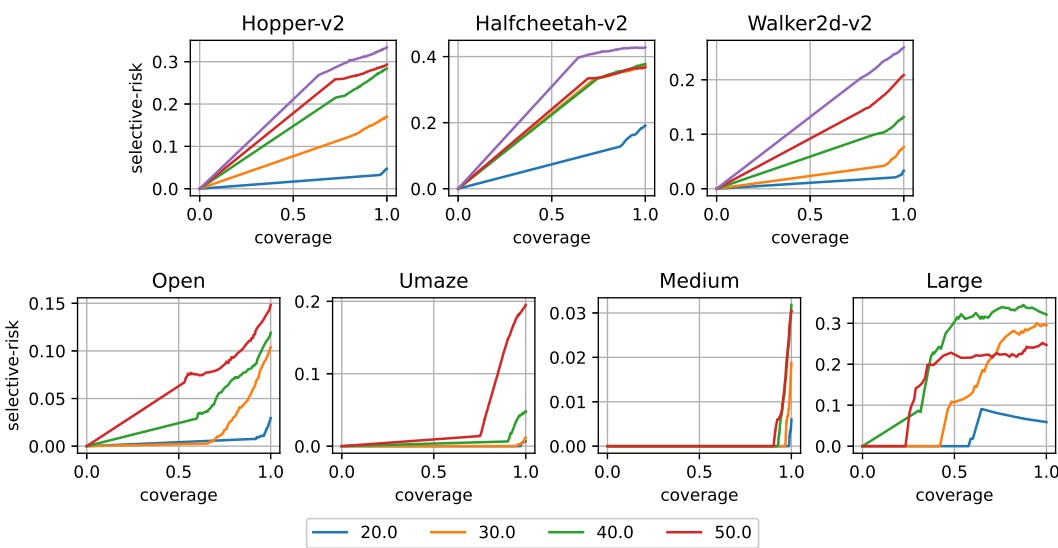

Figure 12: Selective-risk coverage curves for *horizon* ablation in *gym-mujoco* (top-row) and *maze2d* (bottom-row) environments.

