# OpenReview forum: "Offline Policy Comparison with Confidence: Benchmarks and Baselines"
_NeurIPS.cc/2022/Workshop/Offline_RL — Offline RL Workshop NeurIPS 2022_

### Official Review · Reviewer_38o4 · 2022-10-08

**Rating:** 7
**Confidence:** 3

**Review:**

This paper proposed baselines and benchmarks for offline policy comparison with confidence. I believe the authors studied an important topic in offline RL, and their contribution will enlighten and make convenience for the future work along this direction. Therefore, I recommend the acceptance of this paper.

---

### Official Review · Reviewer_CuYE · 2022-10-15
**Well thought of benchmarks for an important problem**

**Rating:** 9
**Confidence:** 4

**Review:**

The authors introduce benchmarks, metrics and baselines for policy ranking approaches that also produce confidence levels. Reliability is a very important problem in offline policy evaluation so I am glad to see this being addressed. The metrics, datasets and baselines proposed make sense to me. I was going to mention that FQE with ensembles would be a good next baseline to try but I think the authors are right that in the context of comparing lots of policies it gets quite computationally expensive. Anyway, all in all a very solid paper that I am happy to see.